# Coumarins into Polyurethanes for Smart and Functional Materials

**DOI:** 10.3390/polym12030630

**Published:** 2020-03-10

**Authors:** José María Cuevas, Rubén Seoane-Rivero, Rodrigo Navarro, Ángel Marcos-Fernández

**Affiliations:** 1GAIKER Technology Centre, Basque Research and Technology Alliance (BRTA), Parque Tecnológico de Bizkaia, edificio 202, E-48170 Zamudio, Spain; seoane@gaiker.es; 2Instituto de Ciencia y Tecnología de Polímeros (CSIC), Juan de la Cierva 3, 28006 Madrid, Spain; amarcos@ictp.csic.es

**Keywords:** coumarin, polyurethane, smart polymers, functional materials

## Abstract

Polyurethanes are of undoubted interest for the scientific community and the industry. Their outstanding versatility from tailor-made structures turns them into major polymers for use in a wide range of different applications. As with other polymers, new, emerging molecules and monomers with specific attributes can provide new functions and capabilities to polyurethanes. Natural and synthetic coumarin and its derivatives are characterised by interesting biological, photophysical and photochemical properties. Then, the polyurethanes can exploit those features of many coumarins which are present in their composition to achieve new functions and performances. This article reviews the developments in the proper use of the special properties of coumarins in polyurethanes to produce functional and smart materials that can be suitable for new specific applications.

## 1. Introduction

Natural and synthetic coumarins consist on a large family of benzo-*α*-pyrones characterised by fused benzene and pyrone rings, where the carbonyl group is located on position 2 related to the oxygen atom within the ring ([Fig polymers-12-00630-ch001]). There are thousands of natural coumarin derivatives from a large number of species of plants, bacteria and fungi. An isolated coumarin molecule from the Tonka bean was first reported in the 1820s, and it was rapidly characterised by an easily recognisable sweet odour similar to than that of new-mown hay. Therefore, since the 1880s, coumarin has been widely used in perfumes, cosmetics and detergents, although mainly as a synthetic compound [1,2,3].

Bourgaud reviewed biosynthesis pathways in regard to natural derivatives, where the different mechanisms remain unresolved or partially clarified as a function of the type of coumarins [4]. One possible classification between different proposed subclassifications of coumarins considers simple coumarins, furanocoumarins, dihydrofurano-coumarins, pyrano-coumarins, phenyl-coumarins and bicoumarins [5,6,7].

From the first described synthesis of the coumarin nucleus in 1868 through the Perkin reaction, synthetical coumarins can be obtained by different methods including the aforementioned Perkin reaction, the Pechman reaction, Reformatsky reaction, Claisen reaction, Witting reaction, the Ring-closing methathesis reaction, Knovenegal reaction, and some other variations incorporating microwave assistance or applying sonochemistry, among others [8,9,10,11,12,13,14,15].

Apart from biological properties and applications in pharmacology as active products, or precursors in new, effective pharmaceuticals, which are reported in different comprehensive reviews [16,17], coumarins are characterised by other interesting properties. Notable photophysical properties of coumarins have turned them into extensive dyes for probing, brightening or labelling, as a function of substitution on the core structure and the interaction with the surrounding media [18,19,20,21,22,23,24]. Besides, coumarin compounds are widely used in the laser industry as laser dyes, as well as being reported as dopants in light-emitting diodes (LEDs and OLEDs) [25,26,27,28,29,30,31,32].

Furthermore, coumarins are photoreactive chromophores that undergo reversible [2π + 2π] cycloaddition reactions upon irradiations at specific wavelengths in the UV-Vis-NIR region [33]. A photodimerisation reaction of coumarin was first reported by Ciamician and Silver in 1902 by submitting a solution of coumarin in alcohol to sunlight for two years [34]. In all cases, the [2π + 2π]-the cycloaddition-based photodimerisation process occurs between two alkenes of molecules with an extended system of conjugation to form a cyclobutane dimer. This photoreaction occurs suprafacially, as a result of a suitable symmetrical interaction between terminal π-lobes on the same face of each involved alkene [35]. As in the case of other similar cromophores, the photochemistry of coumarin derivatives depends strongly upon several parameters which affect the resulted cyclobutane stereoisomers and related obtained photoproducts [36,37]. In particular, coumarin can present four different dimers when cycloaddition occurs, depending on singlet or triplet complexes involved to yield the syn head-to-head dimer, syn head-to-tail dimer, anti head-to-head dimer and the anti head-to-tail dimer ([Fig polymers-12-00630-ch002]) [38,39,40]. Different studies demonstrate that a not very efficient photodimerisation occurs under irradiation of the parent unsubstituted coumarin in solution, even though photoreaction conditions affect it considerably. For example, the polarity of the solvent influences photochemical reactions, considering that nonpolar solvents promote, in general, the obtainment of anti-dimer structures, whereas polar solvents lead to syn dimers. Furthermore, apart from other factors, like concentration or temperature, the substituents or functional groups close to the double bond in the coumarin structure, as well as the presence of host compounds, influence the yielded different dimers [41,42,43,44,45,46].

In the solid state, the reaction of coumarin compounds has also been studied, considering that solid-state photochemical reactions are greatly affected by topological effects and the surrounding crystalline environment [47,48,49,50]. For instance, Moorthy et al. analysed cyclodextrins as host structures for different coumarin derivatives, and their effect on stereochemistry of photodimerisation in comparison to neat coumarin solids and their solutions in a variety of solvents. Their findings demonstrated that those cyclodextrin complexes affected the yield of coumarin dimers, as well as the obtained stereoisomers [51]. These studies were complemented by Brett et al., which performed a more detailed structural analysis in *β*-cyclodextrins as a photochemically inert and non-constraining environment for photodimerisation. Therefore, crystals of the *β*-CD–4,7-dimethylcoumarin inclusion complex, after UV radiation, formed an anti-head-to-tail dimer, and the study demonstrated a complex mix of intermolecular interactions and the spatial fit involved in dimer packing [52].

On the other hand, the photocleavage of coumarin, in both solution and the solid state, has not been so extensively analysed. It is obvious that the photo-scission of coumarin dimers is a cyclobutane-based cleavage reaction, where at least three different types of cleavage modes from potentially five mechanisms have been identified. Furthermore, those cleavage modes for the involved cyclobutane ring can be determined by the chemical structure of the dimers [53,54,55]. Yet not only the mechanism, but also the cleavage efficiency, is affected by starting the photo-dimerised stereoisomer due to balanced stereo forces of phenyl and carbonyl units and charge delocalisation towards the weaker C–C bond, as it was demonstrated by Jiang et al. [56]. Again, the surrounding media and the presence of substituents could enhance the photocleavage process in comparison to the unsubstituted coumarin dimer, although it remains partially unclear as to the involved mechanism [35,57,58,59].

## 2. Coumarins into Polymers

The above described properties make coumarin derivatives interesting candidates to provide tailored properties in polymer materials for a wide range of potential applications. Thus, the scientific community is continuously working to take advantage of the outstanding biological [60], photophysical [61] and photochemical [62] capabilities of coumarin molecules to achieve new functional polymers.

For example, different detailed reviews and books dealing with smart or active polymers collect some information about coumarin-derived materials and associated responsive performances based upon their properties [63,64,65,66,67,68,69,70,71,72]. As described, thus, the presence of coumarin in their composition allows achieving a fluorescent, self-healing or controlled drug delivery performance, among others, in different polymer structures. Trenor et al. reviewed the use of coumarins in polymers until 2004, where the use of these compounds in electro-optical and photoreversible materials, as well in biopolymers, is discussed in-depth [73]. Beyond 2004, many new developments in polymer technology, where coumarin plays a key role to achieve new functions and performance, have emerged. The photophysical properties of new coumarin-derived molecules continue to be used as active dyes in new polymer structures [74]. For example, some very recent reported scientific works deal with analysing the molecular level interaction of coumarin dopant as a foreign molecule at different concentrations within two different polymer matrices [poly(methyl methacrylate) and poly(vinyl alcohol)] [75]. Other works deal with constructing hyperbranched polymer-based light-harvesting structure with coumarin derivatives as the donor and acceptor chromophores [76]. Similarly, photochemical properties provide polymeric materials with controlled drug-delivery capabilities from a coumarin-based phototriggering effect [77,78,79]. These polymers include also light-sensitive polypeptides for controlled structure and properties by light [80].

Even recently, new patterning strategies to control the topographies of surfaces for microfluidic devices can take advantage of coumarin moieties within polydimethylsiloxane. Progressive modification of surface profiles by controlled radiation and the reversible behaviour of coumarin thus open new opportunities for different applications using photo-patterned microsystems (Figure 1) [81].

At the same time, even a new perspective to the coumarin and polymers involves the synthesis of coumarin derivatives using polymer-supported reagents as an effective way for better isolation and purification [82].

In this scenario, polyurethanes (PUs) can also be combined with coumarin compounds as in the case of other polymers. In particular, polyurethanes are a very versatile class of polymers, thanks to their tailor-made features from wide-range, controlled molecular structures. From their discovery in 1937 by Otto Bayer, polyurethanes have emerged as a suitable solution for countless applications based on very diverse building blocks and resulting properties. The core chemistry consists of a urethane group as the main repeating unit in polymer backbone (Scheme 1). The polyaddition reaction of isocyanates and polyols with the controlled nature of those building blocks, particular stoichiometry and reaction sequence, thus determine the achieved properties. Therefore, from a very wide variety of sources and contained chemical groups in both main chain and side chains from which PUs can be created, an extensive range of classes can be developed: rigid, flexible, thermoplastic, thermoset, solvent-borne, or waterborne, among many others. All those options make PUs key components into many different products, such as coatings, paints, elastomers, sealants, foams, fibres, textiles, biomedical devices, etc. [83,84,85].

But even nowadays more sustainable and less toxic alternative pathways for PU synthesis are emerging, not only from using biobased isocyanates, but also from new isocyanate-free routes. For example, the synthesis of phosgene-free and isocyanate-free carbamates, and the subsequent condensation reaction with an alcohol can lead to urethanes. Besides, thermoplastic poly(hydroxyurethane)s can be synthesised by reacting different cyclic carbonates with a variety of amines (Scheme 1) [86].

Regarding coumarin into polyurethanes, the first approach in 1992 dealt with creating optically-active polymers by reacting coumarin-derived diamines with diisocyanate to achieve chiral recognition capabilities [87]. 

Since then, other works focused around combining coumarins, and polyurethanes have increased and led to different results and materials. Therefore, reported polyurethanes have exploited both the photophysical and photochemical properties of different coumarin moieties towards dissimilar objectives. In this sense, the present review intends to present a complete overview of advances in the use of coumarin in polyurethanes, and the wide range of opportunities that can emerge from achieved results.

## 3. Photo-physics of Coumarins in Polyurethanes

The physical properties of the coumarin ring has always aroused high interest for the scientific community, and proof of this is the wide variety of works devoted to this issue. A large number of the photophysical properties of coumarin motif, such as high thermal stability, high quantum yields and large Stokes shifts, have been studied by emission and EPR measurements [88], however in the polymer area the most important properties are related to their high capacity for UV light absorption and fluorescent character. Both properties are described in more detail below.

### 3.1. Effective UV-Light Absorber 

Taking advantage of their high capacity to absorb UV-light, efforts have been devoted to obtaining smart materials that minimise the adverse effects produced by this light on the eye systems. In this sense, Bruin et al. used Coumarin 102 laser dye as an effective UV light absorber in an intraocular lens [89]. Then, a series of transparent aliphatic polyurethanes were synthesised with low molecular weight amine-based polyols as amorphous thermoset resins for ophthalmic applications. Considering mainly the thermal properties and the water uptake by immersion, both circular disks and kerato-prostheses with a tetrakis(2-hydroxypropyl)ethylenediamine-based polyurethane were fabricated, sterilised and implanted in rabbit eyes. The results demonstrated that after one year from implantation, the eye kept clear and the highly crosslinked polyurethane seemed potentially suited to make those kerato-prostheses.

As the UV absorption of artificial intraocular lens must be optimal to avoid the adverse effects of radiation below 400 nm on the retina, chromophores are an important element in polymeric materials [90,91]. Coumarin 102 demonstrated solubility in the polyol component and kept UV absorption after the crosslinking process, unlike other hydroxyl-containing alternative chromophores based on benzophenones and benzotriazoles. Therefore, the polyurethane formulated with just a 0.1% *w*/*w* of this dye was characterised by complete absorption of UV radiation below 450 nm.

### 3.2. Fluorescent Probes 

Pristine coumarin shows little fluorescence behavior because of the lowest excited singlet state of coumarin is an n,π* type [20]. However, the introduction of different substituents to the coumarin ring leads to an electronic character of the first excited state of the π,π* type. This new transition is easily observed in the blue-green region by fluorescence spectroscopy, and has been explored in the production of laser dyes and fluorescent probes. In this sense, Hydroxy-functional coumarins were also used as fluorescent dyes in polyurethane ionomers too. Dicumarol, 4-hydroxycoumarin and 7-hydroxycoumarin were thus reacted with ionised prepolymers based on toluene diisocyanate (TDI), different polyols, 1,4-butane diol and dimethylolpropionic acid (DMPA) as a hydrophilic internal emulsifier [92]. The authors synthesised the different ionomers by a variant of the acetone process [93]. Therefore, after synthesising and neutralising the NCO-terminated polyurethane prepolymers in *N,N*-dimethylacetamide, these were dissolved in acetone, and the coumarin derivatives were introduced as chain extenders ([Fig polymers-12-00630-ch003]), or end capping agents depending on the functionality of the coumarin compounds. Once the water was added and the acetone was evaporated, water-based polyurethane dispersions with fluorescent properties were obtained. In particular, the polymers showed fluorescence at 304 nm, although the quantum yield was not evaluated.

The authors showed that the increase of UV absorbance with coumarin concentration in *N,N*-dimethylacetamide was not linear, due to formed coumarin aggregates. That absorption could also decrease from too large aggregates in the case of a higher content of dicumarol. However, the surface tension of the monofunctional coumarins slightly decreased with concentration, whereas that surface tension increased with dicumarol content due to increased hydrophilic groups being more ordered at the surface of the solvent. Besides, the number average particle size measured by dynamic light scattering increased with the concentration of coumarins. According to the authors, the free volume of ionomer molecules increased due to the intermolecular interaction of hydrophilic groups. However, the attributed increment in particle size with coumarin content was different for the three analysed molecules, due to the dissimilar balance between intramolecular and intermolecular interactions related to the steric effect. Finally, the variations in mechanical properties with coumarin content also were accordingly affected by intermolecular and intramolecular interactions for each coumarin molecule. The tensile strength and elongation increased with the 7-hydroxycoumarin and 4-hydroxycoumarin content. However, the elongation seemed to decrease with the increasing concentration of the difunctional coumarin compound from compact conformation.

Similarly, other authors prepared again a new fluorescent, waterborne, polyurethane dispersion by an acetone process [94]. In this case, the anchoring of 7-amino-4-(trifluoromethyl) coumarin (AFC) into the polymer chains was performed by reacting this chromo-phoric coumarin molecule with anionic NCO-terminated polyurethane prepolymer based on isophorone diisocyanate (IPDI), poly(neopentylglycol adipate) and DMPA. Triethylamine (TEA) acted as the neutralising agent. When the fluorescent intensity of AFC and AFC-modified polyurethane dispersion were compared, this last one showed considerably enhanced fluorescence intensity (>300%). The authors demonstrated that the relatively low fluorescence of AFC in acetone was not attributed to concentration related to self-quenching by comparing fluorescence for different concentrations. Besides, the steric hindrance effect was analysed by dissolving the coumarin molecule in different organic acids. It was observed that the higher molecular size of the acid increased the volume of the formed salt. Therefore, associated steric hindrance hampered the formation of exciplex in AFC, and then the fluorescence intensity increased. The enhanced fluorescence of AFC-modified polyurethane dispersion thus came from a hindered formation of exciplex among AFC moieties by promoted steric hindrance within the polymer. Besides, the light absorption area was increased by the formation of an electrical double surface layer, and the AFC units were shielded from possible quenchers.

An alternative to create fluorescent polyurethanes bearing coumarin was based on creating polyols with fluorescent properties by Huisgen 1,3 dipolar alkyne-functionalised polyether polyol cycloaddition (‘click’ reaction) [95]. Velencoso et al. synthesised an alkyne-functionalised polyether polyol by anionic ring opening polymerisation of propylene oxide (PO) and glycidyl propalgyl ether (GPE) to obtain an alkynyl polyether. Separately, an azide-functional coumarin was synthesised from 4-bromomethyl-7-methoxycoumarin for further linkage with the alkyne group of the polyol in the presence of a copper-based catalyst. In particular, 4-azidomethyl-7-methoxycoumarin was synthesised from the corresponding bromide by nucleophilic substitution with sodium azide in an acetone/acetonitrile solution at 50 °C (Scheme 2).

The preparation of fluorescent polyol was performed by click chemistry with two different copper catalytic systems to establish the efficiency. Although both routes were successful to obtain fluorescent coumarin-functionalised polyols, the best conditions were obtained with copper(I) bromide as our catalyst, *N,N*-diisopropylethylamine as a nitrogen base and tetrahydrofuran (THF) as the solvent. In this last case, the obtained functional polyol achieved a molecular weight (Mn) and a polydispersity of 1410 g·mol^−1^ and 1.6, respectively. Furthermore, the spatial conformation seemed to be notably affected by the introduced coumarin moiety into the polymer backbone.

Subsequently, a rigid PU foam was synthesised with a fraction of the synthesised fluorescent polyol (25% parts by weight), a conventional polyether polyol, methylenediphenyl diisocyanate (MDI), and some additives. Then, the coumarin-containing PU foam was compared with the conventional PU. A characteristic UV-visible absorption peak from the incorporated coumarin function was observed, unlike the PU fabricated with the commercial polyol.

## 4. Photochemistry of Coumarins in Polyurethanes

For a long time, many efforts have been devoted to the preparation of smart functional materials capable of introducing a differential elements leveraging of the photochemical properties induced by the coumarin ring. Although there are works oriented in the biomedical sector, through molecular recognition processes or controlled drug release, much of the effort has focused on the preparation of materials with advanced performance achieved after the UV irradiation process. In the following section, those works focused on various photochemical processes will be highlighted due to the unique photochemical properties of coumarins.

### 4.1. Recognition Ability 

To the best of the authors’ knowledge, Chen et al. [87] prepared optically-active polyurethanes by incorporating molecules derived from coumarin dimers for the first time. The authors considered that ordered conformation and interactions between functional groups are fundamental in chiral recognition. Furthermore, the ring-opening polyaddition of a chiral coumarin dimer can result in optically-active polyamides with efficient resolution to aromatic racemates [55,96].

In this scenario, the different synthesised polyurethanes were evaluated as chiral stationary phases for the resolution of enantiomers having aromatic groups and sites for hydrogen-binding by HPLC. Then, three optically active polyurethanes were synthesised by reacting MDI with amine-functional diastereomers and a diester, respectively (Scheme 3). Those diastereomeric diamines were prepared by reacting previously photodimerised benzophenone (anti head-to-head coumarin dimer) with (R)-(+)-1-phenylethylamine. The potential high recognition ability to aromatic racemates could come from the difference stability of the temporary diastereomeric complexes formed when simultaneously interacting with the racemates. Meanwhile, the optically active diester was obtained by lactonising one of the obtained diamines and performing a ring-opening addition of methanol with that lactonised coumarin stereoisomer.

The chiral stationary phases were prepared by absorbing the polyurethanes on silica gels, which were previously surface treated with dimethoxydiphenylsilane as a phenyl functional coupling agent. Then, the expected chromatographic resolution from interaction sites for chiral recognition was evaluated with a series of racemates containing aromatic groups and sites for hydrogen-binding. From some satisfactory results with racemates containing aromatic groups, the authors proposed a discrimination mechanism of more effective diamine-derived polyurethanes based on simultaneous aromatic stacking and a hydrogen-bonding interaction. Besides, the evaluation of the influence of the polarity of the eluents on recognition capabilities demonstrated a negative effect of increased polarity.

### 4.2. Controlled Drug Delivery 

Coumarin 6 (C6) dye was also used as a fluorescent probe for monitoring the drug entrapping and delivery efficiency of enzyme azo-polyurethane nanoparticles [97]. Coumarin 6 (C6) is frequently used as a fluorescent hydrophobic drug model to trace drug delivery systems via fluorescence spectroscopy [98]. Besides, efficient colon-specific drug delivery systems as effective mechanisms to treat inflammatory bowel disease (IBD) are a challenging objective that is gaining increased importance [99,100,101]. Therefore, Naeem et al. [97] combined a commercial pH-sensitive methacrylate polymer (Eudragit S100) with an enzyme degradable azo-polyurethane to fabricate coumarin-loaded nanoparticles. In particular, the degradable polyurethane containing azo aromatic groups was synthesised via the procedure previously described by Yamaoka et al. in 2000 with *m,m’*-di(hydroxymethyl)azobenzene, IPDI, 1,2-propanediol and poly(ethylene glycol) (PEG) [102]. 

Dual sensitive 244 nm nanoparticles were prepared by a slightly modified oil-in-water emulsion/solvent evaporation technique with an acetone/ethanol cosolvent system. The loading efficiency was determined by fluorescence spectroscopy, as well as the in-vitro releasing capabilities in different pH or in the presence of microbial enzymes. Both in-vitro and in-vivo tests demonstrated that dual-sensitive capsules avoided a too premature drug release. Furthermore, the delivery of the C6 more specifically to the inflamed colon by the azo-reductase enzyme was promoted in comparison to conventional pH-sensitive Eudragit-based particles. 

Additionally, as analysed in the previous section, coumarin-based fluorescent probes in controlled delivery processes with polyurethanes have been reported. Then, photo-responsive polyurethane micellar drug delivery systems have also been studied [103]. Based on previous work developing light cross-linkable and pH de-cross-linkable drug nanocarriers by attaching 7-((4-15 oxopentyl)oxy)-4-methylcoumarin to poly(ethylene-glycol)-poly(hydrazine)aspartamide block polymer [104], active polyurethane micelles were synthesised. A polyurethane micellar structure comprising hydrophobic L-lyisne ethyl diisocyanate (LDI) and hydrophilic poly(ethylene glycol) was designed for drug administration. The key point was to introduce the same coumarin derivative via the hydrazine bond and to crosslink the micelles with 365 nm UV light through a dimerisation process to provide increased stability. Afterwards, folic acid was incorporated via an amination reaction as a model targeting molecule. In this scenario, it could be proposed as to the light-triggered cleavage as a releasing mechanism, but UV not only is hard to penetrate the body, but also it is considerably harmful for tissues. Therefore, pH-responsive hydrazone groups supply the micelles with responsive capabilities at pH values of extra-cellular tumours ([Fig polymers-12-00630-ch004]).

Both pristine and crosslinked micelles were prepared and evaluated considering different aspects, such as drug loading content (DLC), the encapsulation efficiency (EE) and cytotoxicity, as well as its behaviour (size and drug release) at different conditions. On the one hand, the crosslinking process seemed to slightly reduce the average size of spherical micelles, but substantially increased the stability of nanospheres. Unlike original micelles, crosslinked ones kept a practically stable particle size after multiple dilutions. Besides, at acid conditions, despite the potential decomposition of the hydrazine bond that produced the breakage of the micelles, the prolonged times at those acid conditions did not generate any further change.

With regards to cytotoxicity, both neat and crosslinked micelles showed similar and high cell viability. Using Doxorubicin hydrochloride (DOX) as hydrophobic anticancer drug, both the drug loading content and the encapsulation efficiency of light-crosslinked micelles increased up to 55%.

Furthermore, the release profile in response to different pH demonstrated a faster response as the pH decreased, although premature release at slightly acidic conditions was reduced by crosslinked structure. The evaluation of cell uptake and the distribution of DOX with HeLa cells by confocal laser scanning microscopy demonstrated that folic acid played a key role in promoting the specific cell binding and cellular uptake of the crosslinked micelles. Besides, the crosslinked structure of drug-loaded micelles did not disturb the delivery of DOX in vitro, being that the folic acid is an interesting function to enhance the potential target therapy.

### 4.3. Enhancement Mechanical Properties by Irradiation 

In another vein and in relation to photochemically-active polyurethanes able to alter their structure or mechanical properties with light, hydroxy functional coumarin dimers were introduced in the polymer chain already in 1997 [105]. In this approach, 7-acetoxycoumarin and 4-methyl-7-acetoxycoumarin were synthesised from the respective hydroxyl functional coumarin molecules. Both derivatives were irradiated at 350 nm in dichloromethane with benzophenone as the photosensitiser to produce anti-head-to-head 7-hydroxycoumarin dimer and anti-head-to-tail 7-hydroxy-4-methylcoumarin, respectively. The subsequent di-hydroxy functional monomers were produced by hydrolysing them. Afterwards, those difunctional monomers were reacted with the corresponding aliphatic (hexamethylene diisocyanate—HDI) and aromatic diisocyanates (MDI and phenylene diisocyanate—PDI) to create polyurethanes with photodimerised coumarins into the main chain. Once the polyurethanes were prepared, a photocleavage process was performed by irradiating solutions in 1,4-dioxane at 254 nm. The kinetics were monitored by UV spectroscopy. The described equilibrium between photocleavage and photodimerisation under 254 nm made the researchers to trace the photocleavage phenomenon by analysing the absorption change at 315 nm, which corresponds to the B band absorption by the phenyl groups. Polyurethanes with aromatic isocyanates cleaved faster than those with aliphatic ones, according to the shorter time necessary to reach the maximum of absorption. Furthermore, the 4-methyl group of some coumarin derivatives seemed to promote the cleavage of the cyclobutane rings due to the steric repulsion and electron-releasing properties of that methyl substituent [36].

This studio was completed with the repolymerisation of cleaved polyurethanes dissolved in dioxane by the irradiation at 300 or 350 nm. The analysis of changes in absorption with time demonstrated more efficient photodimerisation process at 300 nm from being closer to the maximum absorption of studied photoreactive coumarins. Even though it is true that there was not an analysis of the effect of photolysis and re-photopolymerisation on the chemical–physical properties of those polyurethanes, it seems reasonable to expect important changes in their thermomechanical response. 

Aguirresarobe et al. tested a new the 7-(hydroxyethoxy)-4-methyl coumarin monomer, but as a reactive end-capper for reversible photoinduced extension of waterborne polyurethanes in solid state [106]. The anionic polyurethane dispersions were synthesised by the acetone process. Then, after synthesising the polymer containing hydrophilic groups in acetone, the PU was emulsified in water, and the acetone was removed. In particular, an isocyanate-terminated prepolymer was developed with IPDI, propylene glycol (PPG) and 2-bis (hydroxymethyl) propionic acid (DMPA), which was neutralised with TEA. Subsequently, that prepolymer was extended with 1,4-butanediol (BD), but leaving free isocyanate groups for further reaction with the mono-hydroxy coumarin in the last step. The polyurethanes were designed to contain different proportions of coumarin (1, 5, 10, 15 and 20 wt.%) with the final stochiometric ratio of NCO and OH groups ([Fig polymers-12-00630-ch005]).

The analysis of the particle size of water dispersions demonstrated an outstanding effect of the concentration of coumarin from noncovalent interactions among these end-capping groups. The particle size and the instability of the dispersions increases with the 7-(hydroxyethoxy)-4-methyl coumarin content. In casted solid films, the dimerisation of coumarins at 365 nm reached a maximum degree of 77%, whereas the subsequent cleavage of cyclobutane rings at 254 nm was not complete. The dimerization degree is slightly reduced and the photocleavage dropped substantially as the number of cycles were repeated. The photoreversibilty consequently decreased again from the previously mentioned equilibrium between dimer and coumarin species, as well as from the asymmetric photocleavage reactions that create non-photocrosslinkable species after the opening of the lactone ring [53,106].

The authors of this review came back to 7-(hydroxyethoxy)-4-methyl coumarin as a monomer for preparing photoreactive polyurethanes. This mono-hydroxylated coumarin (5 wt.% and 10 wt.%) was reacted with polycaprolactone (PCL) diol, triol or tetrol to synthesise coumarin end-capped linear and branched polyurethane [107]. However, in this case the polymer structure was controlled to avoid gelation processes that were reported previously in similar structures [108,109]. The study of crystallinity demonstrated that coumarin end-groups could segregate from the polymer matrix in an ordered phase. The photodimerisation of polyurethanes produced an elastomeric material with increased mechanical performance that decreased again with the photo-scission of coumarin moieties (Figure 2). Nevertheless, the polymers irradiated at 313 nm presented an important irreversibility with respect to photocleavage at 254 nm. An optimum reversible photodimerisation/photocleavage response was achieved where it was irradiated with 354 nm lamps for photodimerisation, although some irreversibility with cycles occurred in all cases.

A new coumarin diol was synthesised by the authors [110], 7-(hydroxyethoxy)-4-methyl coumarin was reacted with previously synthesised isopropylidene-2,2-bis-(methoxy)propionic anhydride (DMPAA) to produce an ester. Then, the protecting group was removed to yield the new di-hydroxylated coumarin monomer ([Fig polymers-12-00630-ch006]).

The new coumarin diol was used as the initiator for the synthesis of coumarin-polycaprolactone diols via the ring opening polymerisation of *ε*-caprolactone. Subsequently, an HDI-based polyurethane was synthesised with the resultant mono- and di-substituted polycaprolactone diols containing pendant coumarin groups. The radiation of polyurethane solid films at 354 nm and 254 nm produced the respective photodimerisation and photocleavage reactions. Both UV and Raman spectroscopy showed the same photoreactive conversions, although no direct equivalence was demonstrated between the photodimerisation conversion measured by Raman and the relative crosslinking measured by low-field NMR. This discrepancy was probably associated with the existence of defects and entanglements in the formed network. 

Again, photodimerisation and photo-cleavage showed continuous loss of efficiency, cycle by cycle, being more significant for photo-scission reactions (Figure 3). The photo-crosslinking process reduced the crystallinity of the polyurethane and tough elastomeric material with superior mechanical performance, until similar described materials were obtained.

Salgado et al. analysed the influence of the molecular weight of poly(*ε*-caprolactone) as soft segments on the performance of photoreactive polyurethanes based on the same coumarin diol [111]. A similar procedure developed by Seoane et al. [110] was carried out to synthesise the dihydroxyl coumarin. However, a coumarin- and HDI-based polyurethane prepolymer was produced before further reaction with PCL-diols (Mn = 530 and 2000 g·mol^−1^). The analysed coumarin concentrations were 5 mol%, 15 mol% and 25 mol% for evaluating the different content of hard segment.

The polyurethanes were characterised by the presence of both mono- and bi-substituted coumarin diol. The content of monosubstituted specie increased with the coumarin content (evaluated by ^1^H NMR and ATR-FTIR). Furthermore, the procedure based on the prepolymer seemed to promote bi-substituted species in the structure, as well as higher molecular weights with lower polydispersity when compared to previous authors’ work [110]. 

The photodimerisation degree was promoted by a higher amount of coumarin diol. Besides, the polyurethanes based on shorter PCL soft segments were characterised by higher dimerisation and cleavage yields, probably from the higher mobility of smaller PCL chains.

The coumarin species within the structure affected differently the thermal transitions and the crystallinity depending on the length of soft segments and the coumarin content. The photo- dimerisation process produced some expected and unexpected results depending on the formulation and coumarin content. Then, it was observed from the reduction of crystallinity and increment of *T*_g_ to surprising drops of *T*_g_ in some polyurethanes when photo-crosslinked. Finally, polyurethanes with shorter soft segments showed higher tensile strength, whereas PU based on 2000 g·mol^−1^ PCL were characterised by a higher Young’s modulus. The analysis of mechanical performance demonstrated optimal improvements when a small amount of coumarin was introduced, whereas too high a coumarin content led to detrimental properties.

Salgado et al. also introduced silica nanoparticles to increase the mechanical performance and the thermal stability of those previously synthesised, photo-responsive PUs [112]. Different effects were observed as a function of the macromolecular structure of the PU from different PCL lengths. The effect of nanoparticles on crystallinity was dependent upon the molecular weight of the soft segments and the concentration of nanofillers. These variations coexisted with the effect of the photoreactive processes of coumarin moieties within the structure. Then, different results were obtained from the specific balance between occurring phenomena in each individual case. Anyway, the presence of nanoparticles improved the thermal stability and the mechanical resistance without detriment to the effect of the irradiation process.

Then, an alternative approach was to functionalise silica nanoparticles with a new coumarin derivative and reinforce the same coumarin/PCL-diol-based polyurethanes [113]. The functionalisation was based on coupling an aminosilane to the hydroxylated surface of silica nanoparticles. The reaction of 7-hydroxy-ethoxy-4-methylcoumarin with succinic anhydride produced a new carboxylic-terminated coumarin specie that subsequently reacted with those surface amino groups on silica ([Fig polymers-12-00630-ch007]).

Polyurethane nanocomposites with a 5 mol% ratio of coumarin were prepared by loading 1 and 3 wt.% of unmodified and modified nanoparticles in polymers synthesised with both PCL-diols (530 and 2000 g·mol^−1^). The photodimerisation yields were higher for nanocomposites manufactured with modified silica nanofillers than that for those both non-reinforced and reinforced with unmodified nanoparticles. An improved interaction between the coumarin moieties of the polyurethane matrix and modified reinforcement was demonstrated. However, enhanced crystallinity from modified nanoparticles led to less efficient photo-cleavage materials due to UV affecting more the amorphous phases. 

The performance was again different for short and long chain PCL diols in soft segments due to different phase segregation in each case. Then, the authors concluded that the mechanical performance is only maintained when the 2000 g·mol^−1^ PCL diol is used. Moreover, some improvements in steel corrosion were reported with these coumarin-modified silica nanoparticles.

The authors of this review studied the effect of di-hydroxy-coumarin species in IPDI-based linear polyurethanes within the soft segment and the hard segment, as well as equally distributed between both segments [114]. Both 2,2-bis(hydroxymethyl)propionate of 7-hydroxy-4-methyl-coumarin and 2,2-bis(hydroxymethyl)propionate of 7-hydroxyethoxy-4-methyl-coumarin were synthesised ([Fig polymers-12-00630-ch008]). The subsequent ring opening polymerisation of *ε*-caprolactone with both coumarin diols as initiators to get coumarin-functionalised PCL polyols as soft segments led to different results. The first coumarin species produced transesterification processes between the growing PCL diol and the ester group of the coumarin. In this side reaction, the coumarin ring is excluded from the PCL polymer chain. Therefore, this di-hydroxy-coumarin was discarded for PCL-coumarin soft segments due to generating species with functionality above 2 that could create crosslinked polyurethanes when reacted to isocyanate and chain extender. As an alternative, a second monomer was proposed as an initiator, which produced polycaprolactone diols with the expected structure by ring opening polymerisation, although mono- and di-substituted species coexisted. The polyester functionalised with coumarin motifs was used for the synthesis of new polyurethanes.

The photodimerisation again caused crosslinked polyurethanes with improved mechanical response, i.e., from weak and soft polymers to tough elastomeric materials with outstanding properties. The best performance was achieved when coumarin was within the hard segment. Besides, the photo-scission led to lower mechanical properties from decreased crosslinking, even though limited by lower efficiency that kept considerable toughness in partially cleaved structures (partial reversibility). Anyway, the photodimerisation/photocleavage cycles did not demonstrate any significant effect of location of coumarin on photoreactive kinetics. 

### 4.4. Self-Healing Ability 

In recent years, the design of new materials capable of extending their useful life and improving their safety is becoming important. In this sense, intrinsic self-healing is emerging as a very promising technology. This new technology is based on chemical reactions with a dynamic equilibrium, such as the Diels–Alder reaction or the dimerisation of a coumarin ring [115]. 

Thereupon, following the strategy developed by Ling [116], new light-stimulated, self-healing polyurethanes have recently been introduced using a coumarin dimer as a hard segment chain extender [117]. Then, 7-hydroxy-4-methylcoumarin was photodimerised in solution to form a di-hydroxy coumarin dimer. The polyurethanes were synthesised by reacting an MDI and PTMEG (Polytetramethylene glycol)-based isocyanate terminated prepolymer and the di-functional coumarin dimer. The use of aromatic isocyanate and higher molecular weight chain extender led to enhanced mechanical performance. Nevertheless, a higher fraction of coumarin species involved slight decreases in thermal and mechanical properties, and even elasticity under cyclic loading-unloading tests.

The corresponding self-healing capabilities were evaluated after defining the optimal photoscission/photodimerisation conditions at 254 nm and 365 nm, respectively. Then, previously cut samples were photo-cleaved and -dimerised for the repair process. The self-healing efficiencies were calculated by comparing the cyclic tensile mechanical properties of original and healed materials. Repair efficiencies of the elongation and the Young’s modulus above 90% were achieved, and this skill increased as the coumarin content increased.

Then, Ling et al. worked in taking advantage of the potential photo-crosslinking ability of coumarin molecules to result in polyurethane networks with self-repairing properties [116] The initial approximation used 7-(hydroxyethoxy)-4-methylcoumarin manufactured by converting the aromatic hydroxyl of 7-hydroxyl-4-methylcoumarin into an aliphatic one for further reactivity. This monofunctional coumarin specie was envisaged as a photoreactive pendant group in a polyurethane network. The polymer consisted of trifunctional hexamethylene diisocyanate and a low molecular weight polyethylene glycol ([Fig polymers-12-00630-ch009]). Then, despite the probable partial gelation during the synthesis process, the obtained polyurethane network showed outstanding photo-remendability after mechanical damage. The reversible photodimerisation and photocleavage of the coumarin pendant group on the fractured surface in the solid state led to light-induced healing and the partial recovering of initial mechanical performance. 

The authors monitored the partially reversible photo-crosslinking of film samples by UV-Vis spectroscopy and Raman spectroscopy. The photoreactivity was recorded when irradiated at 350 nm (dimerisation) and 254 nm (cleavage). Partial irreversible crosslinking at a low wavelength (254 nm) during scission was also reported, which could probably come from the involved dynamic equilibrium between photodimerisation and photo-cleavage [53]. 

With previously dimerised polyurethane films, the presence of cleft coumarin species on a fractured surface was demonstrated. Besides, the irradiation of that surface at 254 nm increased the concentration of original coumarin molecules, which promoted the subsequent healing capability throughout photodimerisation under UV radiation. Then, both the qualitative and quantitative analysis of remendability reported outstanding results. The visual inspection of cuts exposed to 350 nm light demonstrated the healing capabilities of the polyurethane-containing coumarin species (Figure 4). Furthermore, the comparison of the tensile strength between original samples and the samples healed by UV illumination after the fracture exhibited healing efficiencies above 70% under the best conditions during the first test cycle. This efficiency decreased with the consecutive healing cycles from the progressive reduction of reversible dimerisation, as well as from the potential misalignment of the fractured surfaces during testing. These healing capabilities were even demonstrated under sunlight illumination, but lower efficiencies were achieved.

Even again with the monofunctional 7-(hydroxyethoxy)-4-methyl coumarin, the effect of the molecular weight of the PEG segments (Mw = 200, 400 and 800 g·mol^−1^) on the healing capabilities was studied [109]. The higher molecular weight and the corresponding lower *T*_g_ (glass transition temperature) of the PEG segments promoted the dimerisation and decrosslinking processes under UV at room temperature. However, the healing efficiencies were characterised by an equilibrium between the concentration and the photoreactivity of coumarin moieties within the polyurethane network. The polyurethane with the 200 g·mol^−1^ PEG showed a higher concentration of coumarins and related healing efficiency for the first healing cycle. 

On the contrary, the 800 g·mol^−1^ PEG provided more photoreactive reversibility for consecutive cycles, but with a lower absolute healing capacity. Therefore, the authors suggested the medium molecular weight PEG as the more effective soft segment for achieving a balance self-repairing performance in this PU network.

Then, as a step forward to obtain self-healing polyurethanes, the same authors introduced dihydroxyl coumarin as a chain extender, which was a more suitable alternative to the monofunctional photoreactive moiety. According to the authors, the difunctional coumarin allowed us to overcome the gelation-related drawbacks of the previous approach. Besides, it enabled easier the adjustment of the formulation without changing the molecular weight of the polyol [118]. The authors thus synthesised, 5,7-bis(2-hydroxyethoxy)-4-methyl coumarin, reacting 5,7-dihydroxy-4-methyl coumarin [108] with 2-bromoethanol. A series of photoreactive polyurethanes were produced by reacting IPDI with dihydroxyl coumarin, and subsequently the isocyanate terminated the prepolymer with different molecular weight PEGs with the corresponding molar ratios ([Fig polymers-12-00630-ch010]).

The analysis of photoreactivity demonstrated the reversible crosslinking of the polymers by the photodimerisation and photoscission of coumarin species into the macromolecular network. However, the photoreversibility decayed again through the cycles due to an asymmetric breakage of dimers during photocleavage [53]. Besides, according to the authors, the achieved reversibility into the synthesised polyurethanes was not mainly affected by the concentration of coumarin molecules, but by the enhanced mobility of rubbery domains in phase-separated structures.

The optimal photo-remendability based again on the presence of cleft coumarin on the fractured surface was achieved with the fractured surface exposed 1 min at 254 nm and then 90 min at 350 nm to promote the photodimerisation-based healing. The tensile strength restoration was affected by surface-localised viscoelastic flow and mainly by optimal photochemical reactions. Thus, a maximum of 100% of healing efficiency was achieved for a polyurethane based on the higher molecular weight PEG and a 0.25 molar ratio of di-functional coumarin from higher mobility (lower *T*_g_ and higher molecular length between crosslinks).

Recently, various government organisations such as the European Union, according to the Kyoto protocol, have issued a series of measures to reduce greenhouse gas emissions from the material production stage to the end user. To fulfil this agenda, bioplastics have emerged as a very promising alternative [119]. In these polymers, there is a significant percentage derived from the biomass (plants, animals, fungi or bacteria). Following this new trend, Badri et al. [120] have presented a series of biobased, self-healing polyurethanes. In their chemical structure, these authors have used as a soft segment a polyol obtained from palm kernel oil, and the coumarin units are based on natural derivatives such as Esculin.

### 4.5. Shape Memory Polymers 

That of shape memory polymers is a hot topic that has emerged with great interest in recent years; in fact it has been critically reviewed in depth by various authors [121,122]. These smart materials present the capability of recovering their initial shape from a temporary shape under the action of an external stimulus, such as pressure, light, heat, pH, etc. 

Following the main line of the present review, photochemical activation is probably one of the most used mechanisms in this issue, due to a wide variety of available photoreactive molecules and spatial control, allowing us to selectively irradiate some regions against others. Since the implementation of this methodology, the first efforts focused on preparing polyurethanes carrying other photoreactive probes, such as anthracenes [123,124], cinnamates [125] and azo-benzenes [126]. On the other hand, coumarins have been inserted into other polymer systems such as PVA [127] or polyesters [128,129,130], achieving very encouraging results. However, it is not until the beginning of 2019 where the chemistry of coumarins was carried out within the polyurethane matrix in order to obtain shape memory materials [131,132]. In both cases, the coumarin insertion was carried out by the ‘click’ reaction. In order to achieve this shape memory effect, it was necessary to control the chemical composition of the polyurethane and the molecular weight of the soft segment, since it was required that both the soft and hard segment crystallise. In conclusion, these authors presented a new approach for achieving a UV/heat dual responsive triple-shape memory effect in photo-responsive coumarin-containing polyurethanes. The recovery ratios of temporal shapes could be easily adjusted by only tuning the crosslinking intensity, heating and time. Meanwhile, the authors also found that one of these temporary shapes was very stable, and was used as a complex permanent shape for the thermally sensitive shape memory effect. The potential of these polyurethanes to create stable and tunable original shapes will be beneficial for materials with permanent shape reconfigurability in different sectors.

## 5. Other Properties of Coumarin Additives in Polyurethanes

Taking advantage of other potential properties of coumarin derivatives, El-Wahab and colleagues have studied the effect of a new series of coumarins as functional additives in a polyurethane coating [133,134]. Their first approach was based on creating a hybrid antimicrobial structure from coumarin and a thiazole ring, which are identified biologically active compounds. In particular, 2-(2-amino-1,3-thiazol-4-yl)-3H-benzo[f]chromen-3-one was synthesised, and it was physically added to a polyurethane coating formulated with refined sunflower oil, glycerol, pentaerythritol, toluene diisocyanate and turpentine, among other components. The antimicrobial activity of the modified polyurethane was evaluated with different target bacteria and fungi, being more effective against Gram-positive bacteria (G+). Then, the improved performance was associated to the presence of the two functional groups within the molecule, i.e., the coumarin ring and thiazole ring [133]. 

More recently, from those previous results, new coumarin and benzocoumarin derivatives incorporating a thiazole ring were synthesised and evaluated again as potential antibacterial additives, but also as flame retardant and corrosion inhibitor compounds [134]. [Fig polymers-12-00630-ch011] shows the chemical structure of these modified coumarins. These coumarins coupled with a thiazole ring were synthesised from previous bromide intermediates based on coumarin and benzocoumarin. After preparing 3-acetocumarin and 3-acetobenzocoumarin from salicylaldehyde and 1-hydroxy-2-naphtaldehyde, respectively, the intermediates were synthesised by treating aceto-derivatives with bromine. Subsequently, those bromine functional compounds reacted with thioacetamide or 2-(4-fluorobenzylidene) hydrazine carbothionamide to obtain the respective new coumarins.

Based again on the reported activity against the bacteria and fungi of coumarin and its derivatives, synthesised molecules were incorporated as functional additives in the same polyurethane coating described above. The antimicrobial activity of the coatings was evaluated with a ratio of coumarin derivatives of 0.5 wt.%, 1.0 wt.% and 1.5 wt.%. Compared to neat control polyurethane formulation, the addition of coumarins led to improved antimicrobial activity from the biologically active coumarin ring and thiazole ring [135,136]. On the one hand, the higher the concentration of coumarin derivatives, the better the effectiveness against the target microorganisms (*S. Pneumoniae* and *B. Subtilis* as Gram-positive bacteria; *P. Aeruginosa* and *E. Coli* as Gram-negative bacteria; and *A. Fumigatus* and *G. Candidum* as fungi). On the other hand, the activity of 2-(2-(2-(4-fluorobenzylidene)hydrazinyl)thiazol-4-yl)-3H-benzo[f]chromen-3-one was higher than that of 2-(2-mthylthiazol-4-yl)-3H-benzo[f]chromen-3-one due to the fluorine substituent, being the 3-(2-methylthiazol-4-yl)-2-H-chromen-2-one derivative the less effective from lower aromaticity than benzo-derivatives. Besides, achieved anti-microbial function was considerable against Gram-positive bacteria, whereas it was just moderate against Gram-negative and mild against fungi.

The flame retardant properties were evaluated by measuring the limiting oxygen index (LOI), which it increases with lower flammability of the plastic material. The LOI value of the polyurethane coating linearly increased with the content of coumarin derivatives for the same 0.5 wt.%, 1.0 wt.% and 1.5 wt.% concentrations, and it was better than that of the neat reference in all the cases. The thiazole ring provided flame retardancy from a synergy effect between nitrogen and sulphur content, although the other substituents promoted the achieved improvement. Then, 2-(2-(2-(4-fluorobenzylidene)hydrazinyl)thiazol-4-yl)-3H-benzo[f]chromen-3-one became the most efficient molecule from the halogen atom and the three nitrogen atoms in the structure. The formulation with a 1.5 wt.% of this additive, thus, doubled the achieved LOI value if compared to the blank polyurethane coating. 

Potential improvement in anticorrosive properties from coumarin additives was studied by evaluating blistering, scribe failure and the degree of rusting on painted steel panels after performing a salt spray (fog) test for 500 h of exposure. Organic heterocyclic compounds are considered organic structures with some effective corrosion inhibition properties [137]. Therefore, synthesised coumarin derivatives were expected to be alternative organic inhibitors for corrosion protection. In this case, the authors evaluated the polyurethane coatings modified with a 1.5 wt.% loading of the three respective coumarin compounds, and the fluoro-derivative, with higher presence of heteroatoms and aromatic rings, was again the most efficient additive.

## 6. Conclusions

Polyurethanes are a very versatile tool to create unique materials with a wide range of properties from tailor made structures and compositions. The incorporation of coumarins into polyurethanes as functional molecules opens a window on new functions and opportunities. The distinctive photophysical and photochemical properties of natural and synthesised coumarin species lead to diverse polymeric materials with extensive performance.

This review summarises the potential applications of coumarins and emphasises the great increment of their use into polyurethane materials. A comprehensive description of the diverse achieved properties and functionalities in polyurethanes allows us to update this field with all relevant related activity. Besides, it encourages the polymer scientist to expand future possible combinations. The possibilities to new functional polyurethanes could be endless.

Different new coumarin molecules with different reactive groups arise as potential chemicals to provide polyurethanes and any other organic material with different and useful responsible characteristics for the future.

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
