# Peer review of "Coumarins into Polyurethanes for Smart and Functional Materials"

_polymers, 2020, doi:10.3390/polym12030630_

Round 1
Reviewer 1 Report
Comments to the Author
This work reported polyurethanes are of undoubted interest for the scientific community and the industry. Their outstanding versatility from tailor‐made structure turns them into major polymers for use in a wide range of different applications. However, the paper have been carefully discussed. Minor revision should be done before publication. The following comments are listed for improvement of the paper:
- In the paper, it has to refer to more relevant literatures on coumarin based-polyurethane of smart and functional polymers in recent years, such as ”Synthesis of triple shape memory polyurethanes by introducing photo-responsive coumarin units into the crystalline soft segment. Materials Today: Proceedings, 16, 1507-1511.” and ”Photo-activated self-healing bio-based polyurethanes. Industrial Crops and Products, 140, 111613.” etc.
- It is necessary to confirm the literatures in detail (such as 21. J. Org. Chem. 26. J. Fluoresc. 30. 33 etc., and the words should be italic, and in 116. the name should be confirmed.)
- In Line 532 ‐ Please confirm the word of caprolactone.
- In Line 552-553 ”Recently, Wang et col. presented new light‐stimulated self‐healing polyurethanes based on the strategy reported by Ling [105] but incorporating a previously photo‐dimerized 7‐hydroxy‐4‐ methylcoumarin into the hard segment as a chain extender [117].” This sentence may be revised to more better.
- In Line 568, El‐Wahab and colleagues should be corrected to El‐
- In Line 116, the word of kerathoprostheses should be corrected to keratoprostheses.
Author Response
In the paper, it has to refer to more relevant literatures on coumarin based-polyurethane of smart and functional polymers in recent years, such as ”Synthesis of triple shape memory polyurethanes by introducing photo-responsive coumarin units into the crystalline soft segment. Materials Today: Proceedings, 16, 1507-1511.” and ”Photo-activated self-healing bio-based polyurethanes. Industrial Crops and Products, 140, 111613.” etc.
Following the recommendation proposed by the reviewer, these mentioned papers and additionally further works have been conveniently introduced into the main text. To facilitate the search and verification, the new references commented by the reviewer correspond to: 132 and 120 respectively.
It is necessary to confirm the literatures in detail (such as 21. J. Org. Chem. 26. J. Fluoresc. 30. 33 etc., and the words should be italic, and in 116. the name should be confirmed.)
Thanks for this observation. All references have been reviewed and updated, ensuring that all of them follow the appropriate format of the journal.
In Line 532 ‐ Please confirm the word of caprolactone.
Thanks for the correction. The word has been corrected.
In Line 552-553 ”Recently, Wang et col. presented new light‐stimulated self‐healing polyurethanes based on the strategy reported by Ling [105] but incorporating a previously photo‐dimerized 7‐hydroxy‐4‐ methylcoumarin into the hard segment as a chain extender [117].” This sentence may be revised to more better.
Following the recommendation by the reviewer, in the manuscript we have included a new paragraph to clarify the concept. The new paragraph appears in line 511-513. “Thereupon, following the strategy developed by Ling [116], new light-stimulated self-healing polyurethanes have recently been introduced using a coumarin dimer as a hard segment chain extender.[117]”
In Line 568, El‐Wahab and colleagues should be corrected to El‐
Thank you, we have corrected the mistake.
In Line 116, the word of kerathoprostheses should be corrected to keratoprostheses.
Thanks for this observation, the spelling error has been corrected in line 174
Reviewer 2 Report
- Putting the coumarins into polyurethane for smart and functional materials is relative novel and rarely reported. It will be interesting to more readers. The author summarized the photochemical and photophysical properties of PU containing coumains. However, the whole paper shows some not clear, logic. Therefore, I suggest the author rewrite it totally. The author can give main catalogue such as: 1.the history of coumains and inserting coumains into PUs. 2. The synthesis route and kinds of PU containing coumarins; 3. The photochemical and photophysical properties of functional PUs 4. The application and function of PUs.
- There should be big title followed by subtiltle, in order to explain clear. for example, 2.1 the self-healing; 2.2 flame retardant properties. 2.3 anticorrosive properties
- The author can divide the paper into two parts, however, photophysical and photochemical parts do not separate clear, I suggest the paper can be separated into synthesis, purposes of coumarins and PU, properties. under primary title, there should be second and third title to describe clearly.
- The format reference of 102, 110, 111 should be conform to others.
- Many references are old which should be updated.
Author Response
relevant published aspects where the properties of coumarins converge within a polyurethane matrix. It is well known that coumarin rings have always attracted a lot of scientific interest due to their photochemical and photophysical properties. Therefore, it is logical to consider that both criteria are the key elements for a correct structuring of the review. In fact, we have found a paragraph that is in line with our criteria and can be found in the following article (DOI: 10.1039/c6pp00399k), where it is written: “Coumarin and its derivatives have been subjected to photochemical and photophysical studies because of their wide range of applications, such as laser dyes and fluorescent labels, as well as the molecular systems established as important photosensitizers for both in vitro and in vivo systems”.
However, we also consider that the reviewer is right in proposing the incorporation of sub-sections that allow to explain in detail the data collected, in order to facilitate their understanding. Therefore, the sections related to photophysical and photochemical properties have been subdivided into small sections highlighting the most important applications included in each section. In this way, we think that the current structure is more relevant and has its own and differentiating elements compared to other old coumarin reviews. Therefore, we hope that with this new division a good compromise will be reached between the structure proposed by the reviewer and the current structure presented.
The format reference of 102, 110, 111 should be conform to others.
Thanks again for this observation. As we have previously commented to reviewer 1, the references have been reviewed and updated, ensuring that all of them follow the appropriate format of the journal.
Round 2
Reviewer 2 Report
The author has corrected the manucript, so it can be published .However, attention should be paied attention to typo writting, such as in line 158, "has", in line 182, "of", in line 619, "withih" should be altered.